# Association of Sleep Duration with Hyperuricemia in Chinese Adults: A Prospective Longitudinal Study

**DOI:** 10.3390/ijerph19138105

**Published:** 2022-07-01

**Authors:** Huan Yu, Kexiang Shi, Haiming Yang, Dianjianyi Sun, Jun Lv, Yuan Ma, Sailimai Man, Jianchun Yin, Bo Wang, Canqing Yu, Liming Li

**Affiliations:** 1Department of Epidemiology & Biostatistics, School of Public Health, Peking University, Beijing 100191, China; yuhh@pku.edu.cn (H.Y.); shikexiang@pku.edu.cn (K.S.); heidiyang@bjmu.edu.cn (H.Y.); dsun1@bjmu.edu.cn (D.S.); epi.lvjun@vip.163.com (J.L.); sailimai.man@meinianresearch.com (S.M.); lmleeph@vip.163.com (L.L.); 2Peking University Health Science Center, Meinian Public Health Institute, Beijing 100191, China; yuan.ma@meinianresearch.com; 3Peking University Center for Public Health and Epidemic Preparedness & Response, Beijing 100191, China; 4Key Laboratory of Molecular Cardiovascular Sciences, Peking University, Ministry of Education, Beijing 100191, China; 5Meinian Institute of Health, Beijing 100191, China; 6MJ Health Care Group, Shanghai 200041, China; yinjc@health-100.cn

**Keywords:** hyperuricemia, sleep duration, longitudinal study, epidemiology

## Abstract

Little is known about the association of sleep duration with hyperuricemia. Especially lacking is evidence from longitudinal studies. Based on the MJ Health Examination Database in Beijing, China, a prospective study was designed. Participants were classed into short, normal, and long groups by sleep duration. The Cox regression model was used to estimate the hazard risk of hyperuricemia for short or long sleep duration compared with the normal group after adjusting for potential confounders. During a median 3.08 years follow-up, 4868 (14.31%) incident hyperuricemia events were documented among 34,025 participants with a crude incidence rate of 39.49 per 1000 persons. Years after adjusting for potential confounders, a 7% higher risk of hyperuricemia in the short sleep duration group (<7 h, 95% confidence interval: 1.01–1.14) and a 15% lower risk in the long sleep duration group (≥8 h, 95%CI: 0.74–0.97) were found compared with the normal group (7–8 h) (*p* for trend < 0.001). Nevertheless, the association of the short sleep duration group was marginally significant after further adjustment of the count of white blood cells (hazard ratio: 1.07, 95%CI: 1.00–1.13). Sleep duration was inversely associated with hyperuricemia, which highlights the public health significance of sufficient sleep duration for preventing hyperuricemia.

## 1. Introduction

Hyperuricemia is considered to be the primary step in the occurrence and development of gout and uric acid calculi, and is strongly correlated with various chronic metabolic diseases, such as gout, hypertension, type 2 diabetes, lipid metabolism disorders, chronic kidney disease, heart failure, and stroke [1,2,3,4,5,6,7,8]. As an inflammatory metabolite, uric acid (UA) is involved in the inflammatory response and activation of the innate immune, and induces the development of diseases [6,9,10]. However, the prevalence of hyperuricemia varies across countries, from 20% in the United States [11] to 6.4% in China [12]. At the same time, the incidence and prevalence of gout and hyperuricemia are constantly increasing around the world, causing a heavier burden and health loss [13]. Therefore, the discovery and control of risk factors for hyperuricemia has become a critical public health issue.

Sleep duration reflects sleep situations and plays an important role in health. Sleep that is both too short and too long sleep has been demonstrated to be related to various diseases and mortality [14,15,16,17]. However, with the current socio-economic development, about one-third of people sleep less than 7 h, one-tenth less than 5 h, and 8% sleep more than 9 h [18].

Existing evidence has shown sleep duration was related to inflammatory molecules [19,20,21] and metabolic diseases. However, there is still a lack of research on the relationship between sleep duration and uric acid levels. A recent cross-sectional study showed a U-shaped relationship between sleep duration and uric acid levels in Korean women [22]. The risk for higher serum UA increased 54% in the short sleep duration group and 94% in long group, compared with that in participants who slept for 7–8 h. Moreover, evidence from an elderly Mediterranean population showed an inverse association between sleep duration and serum UA concentrations [23]. Nevertheless, no longitudinal studies with a large sample size have been reported so far in this field to confirm the association. Our study is a prospective longitudinal study with a large sample to assess the association between sleep duration and hyperuricemia in Chinese adults. It would fill the gap in this field and contribute to public health research on the prevention of hyperuricemia.

## 2. Materials and Methods

### 2.1. Study Population

This longitudinal study aimed to explore the association of sleep duration with hyperuricemia among Chinese adults. This study was based on the MJ Health Examination Database in Beijing, China. The details of the database can be found online [24]. We included participants over 18 without prevalent hyperuricemia at baseline that had two or more examinations from 1 January 2008 to 31 December 2018. We excluded subjects with tumors, cardiovascular diseases, kidney diseases, mental illness, or those taking psychotropic drugs or sleeping pills. In addition, we also excluded participants with missing key covariables, including sleep duration, Serum UA, age, sex, and Body Mass Index (BMI). Finally, 34,025 participants remained in the study (Figure 1).

All participants completed a standardized self-administered questionnaire at each visit to collect their demographic information, lifestyle habits, medication status, and disease history. The physical and laboratory examinations were performed at the MJ Medical Examination Center with medical diagnostic qualifications. The project was approved by the Peking University biological and medical ethics committee (IRB0001052-19077), with written informed consent obtained from each participant.

### 2.2. Assessment of Sleep Duration and Hyperuricemia

The sleep duration was obtained through a unified self-administrated questionnaire. All participants were asked the following question about sleep duration: “On average, how many hours do you sleep per day”. There were five options for the question: (1) <6 h; (2) 6–7 h; (3) 7–8 h; (4) 8–9 h; and (5) ≥9 h. Participants were categorized into three groups: 7–8 h as the reference group, <7 h as the short sleep duration group, and ≥8 h as the long sleep duration group, which was consistent with previous studies [21]. Serum UA was measured using Roche Cobas C 501 biochemical analyzer. Hyperuricemia was defined as UA ≥ 420 μmol/L in males or UA ≥ 360 μmol/L in females [25,26].

### 2.3. Covariates

The questionnaire collected demographic characteristics (i.e., age, sex, marital status, income, education status), lifestyle behavior (i.e., alcohol drinking, cigarette smoking, leisure-time physical activity), and medical conditions. Marital status was classified into two groups: married and the others (including never married, divorced, and widowed). Personal annual income was identified as three levels: CNY < 0.1 million (Chinese Yuan), CNY 0.1–0.2 million, CNY > 0.2 million per year. The education attainment was divided into three categories: bachelor’s degree or lower, master’s degree, and doctoral degree. Smoking status, drinking status, and leisure-time physical activity status were dichotomized into yes or no.

All biochemical markers, including serum creatinine (SCR), high-density lipoprotein cholesterol (HDL-C), triglycerides (TG), blood glucose, and amount of white blood cells (WBC), were measured using a Roche Cobas C 501 biochemical analyzer. Physical examinations, such as height, weight, abdominal circumference, and blood pressure (BP), were performed by trained physicians. BMI was calculated as weight in kilograms divided by height squared in meters, and then dichotomized as a binary variable with a cut point of 24 kg/m^2^ [27]. Abdominal obesity was defined as abdominal circumference >90 cm in males or >85 cm in females [28]. BP was measured with a regular mercury sphygmomanometer on the right arm after resting for at least five minutes. Hypertension was determined as systolic blood pressure ≥140 or diastolic blood pressure ≥90, including the use of anti-hypertension medication or self-reported history of hypertension [29]. Fasting serum HDL-C ≤ 1.04 mmol/L indicated low high-density lipoprotein, and fasting serum TG ≥ 1.70 mmol/L was classified as high triglycerides [28]. Diabetes was determined when fasting serum glucose ≥ 7.0 mmol/L or glycosylated hemoglobin ≥ 6.5%, including use of hypoglycemic drugs and self-reported history of diabetes [30]. WBC classification was described as a binary variable including a low (<7 × 10^9^/L) and high level (≥7 × 10^9^/L).

### 2.4. Statistical Analysis

The distribution of continuous variables was tested using the Shapiro–Wilk normality test. Continuous variables with normal distribution were described as means and standard deviations (SD). Other continuous variables used medians and interquartile ranges. Categorical variables were described as frequencies and percentages. The characteristics at baseline were compared in groups according to sleep duration by the chi-square test for categorical variables, analysis of variance (ANOVA) test for continuous variables with homogeneous variance, and the Kruskal–Wallis test for continuous variables without homogeneous variance. We assigned a separate group for the missing values in categorical variables.

The Cox proportional hazard regression model was used to calculate the hazard ratio (HR) and 95% confidence interval (CI). Survival time was defined as the time difference from the first visit to the examination which documented hyperuricemia or taking UA-lowering drugs for the first time, or the last visit, whenever came first. The incidence of hyperuricemia (per 1000 person-years) was calculated according to the sleep duration category. Four models were estimated: Model 1: Only adjusted for sex and age; Model 2: Additionally adjusted for education level, personal annual income, marital status, smoking status, drinking status, and leisure-time physical activity status based on Model 1; Model 3: was the fully adjusted model in this research, which additionally adjusted for BMI, abdominal circumference, hypertension, diabetes, triglycerides, and high-density lipoprotein cholesterol based on Model (2) and (4); and Model 4: To test potential intermediate factors by inflammation, we further adjusted the amount of white blood cells based on Model 3. To access the linear trend between sleep and hyperuricemia, we also fitted restricted cubic splines with 5 knots to flexibly model the association. To further explore the modification effects by different covariates, subgroup analysis was also conducted.

In addition, sensitivity analyses based on the fully adjusted model were performed as follows: (1) Re-run the cox proportional risk regression analysis by excluding participants with missing data; (2) Use the five-category sleep duration to test the dose–response relationship between sleep duration and hyperuricemia. Two-sides *p* < 0.05 were considered as statistically significant, and all analyses were completed using SAS statistical software version 9.4.

## 3. Results

### 3.1. Baseline Characteristics

At baseline, the mean age was 38.31 ± 9.59 years old and 54.38% of participants were women. A total of 19420 (57.08%) participants reported short sleep duration, and 2301 (6.76%) participants were in the long sleep group. The baseline characteristics were shown by different sleep duration groups in Table 1. Compared with other groups, long sleepers were most likely to be women, younger, to report the lowest education attainment and annual income, and they had the lowest proportion in drinking, leisure-time physical activity, overweight or obesity, abdominal obesity, and low HDL-C (*p* for difference < 0.05 for each) (Table 1).

### 3.2. Association between Sleep Duration and Hyperuricemia

With the follow-up of a median 3.08 (interquartile range: 1.92–5.17) years, 4868 incident hyperuricemia were documented, with a total crude incidence 39.49 per 1000 person-years.

After adjusting for demographic factors, habits, and metabolism-related covariates (Model 3), HR was 1.07 (95%CI: 1.01–1.14) for short sleep duration and 0.85 (95%CI: 0.74–0.97) for long sleep duration. Adding adjustment of white blood cell amount, HR was 1.07 (95%CI: 1.00–1.13) for short sleep duration with marginal significance and 0.84 (95%CI: 0.74–0.97) for long sleep duration in Model 4. All four models showed a significant inverse trend (*p* for trend < 0.001) (Table 2). No significant nonlinearity was detected based on restricted cubic spline (Figure 2, *p* for nonlinearity > 0.05) In the additional sensitivity analyses, the results remained unchanged (shown in Table A1). In the five-category sleep duration analysis, a significant inverse trend (*p* for trend < 0.001) was also observed in all four models (shown in Table A2).

The associations were similar across different levels of covariates. Only the interaction between sleep duration and BMI was significant (*p* for interaction = 0.035) (Table 3). A 16.3% higher risk of hyperuricemia for short sleep individuals with normal weight. A 19.9% lower risk of hyperuricemia for those who reported long sleep duration and overweight or obesity was found.

## 4. Discussion

In this prospective study, a dose–response effect of sleep duration was observed on hyperuricemia in Chinese adults. Short sleep duration was associated with a 7% higher risk of hyperuricemia, while a long sleep duration showed a 15% lower risk. The findings highlighted that extending sleep duration might be a potential public health strategy to prevent hyperuricemia.

Sleep loss is still a global problem, and there is still a lack of corresponding policies worldwide. In this study, only 36.16% of participants had a recommended sleep duration, and over one-half of participants slept less than 7 h. Our results are in line with several previous studies that estimated the association of sleep duration with hyperuricemia. Consistent with previous cross-sectional studies [22,23,31], short sleep duration was considered as a risk factor for hyperuricemia in this study. Appropriate public health actions should be taken to increase the sleep time of the population to reduce the damage to health caused by sleep that is too short, especially among subgroups with a low HDL-C, a high TG, or those who are not overweight.

This study provides further evidence that sleep duration is independently associated with hyperuricemia after adjusting for known risk factors and potential confounders. Unlike a U-shaped curve reported in Korean women [22], our findings indicated that longer sleep (>8 h) could also reduce the incidence of hyperuricemia significantly, which was consistent with previous studies [23,32]. In this prospective longitudinal study, sufficient samples and prospective study design avoided reverse causality by chronological order. Interestingly, although we found that the participants who had a longer sleep duration had a lower risk of hyperuricemia, previous studies showed that long sleep was detrimental to health [33,34,35]. In addition, studies have shown that a low level of uric acid was a risk factor for Parkinson’s disease [32] and Alzheimer’s disease [36]. Thus, the choice of sleep duration needs to comprehensively consider the other problems caused by a long sleep duration.

Two underlying mechanisms could explain our findings of the inverse association of sleep duration with hyperuricemia. One possible way for sleep duration to alter metabolism might be by affecting levels of catecholamine. Catecholamine is an essential neuroactive transmitter, facilitating the breakdown of nucleotides, leading to more production of endogenous uric acid [37]. The activity of locus coeruleus catecholamine was reported to decrease during sleeping, which would reduce the level of uric acid [38]. Moreover, animal models have proved that catecholamines play an important role in the occurrence of hyperuricemia [39]. The other pathway might be related to inflammatory reactions. Studies have shown that sleep duration substantially affected inflammatory mediators, inducing a series of chronic inflammatory diseases [21,40,41]. As an activator of the immune system [10], uric acid enrolls in the body’s inflammatory response. Previous studies showed that the number of white blood cells was a potential intermediate factor between sleep duration and hyperuricemia [23,42]. After adjusting the amount of white blood cells in this study, the association between short sleep duration and hyperuricemia was no longer significant, which supported the hypothesis. Moreover, interactions among metabolites should be noticed. Cicero et al. found that a high serum uric acid level and a high serum LDL-C level would have an interaction on incidence of hypertension [43], which further proves the complexity of interactions among metabolites. In the subgroup analysis, BMI was found to have an interaction between sleep duration and hyperuricemia. This might be related to the different metabolic status of individuals with different BMI level [44]. Notably, short sleep duration is associated with a 16% increase in risk for hyperuricemia in the participants with a BMI < 24, which provides insights into screening high-risk populations and protection of susceptible people.

To the best of our knowledge, the present study is the first population-based prospective study which included a large sample size to study the relationship between sleep duration and hyperuricemia. We conducted a comprehensive analysis by adjusting potential risk factors and confounders, including comprehensive laboratory markers, which enabled us with to examine the possible mechanism of the association.

However, some limitations should be mentioned. First, the participants enrolled in the present study were from Beijing, with higher education attainment and annual income, but a relatively lower percentage of long sleepers as compared to the other studies, which limited the extrapolation of our findings. Second, the follow-up time was calculated from each health check-up visit, hence diagnosis time could be delayed, leading to underestimation of the association. Third, some covariates were missing. However, the sensitivity analysis showed the robustness of our findings using a simple imputation. In addition, although the sleep duration related questions were similar to those in previous studies [18,45], our questionnaires were not validated. Further validation of the questionnaires is needed in future research.

## 5. Conclusions

Sleep duration was inversely associated with hyperuricemia, which indicated that short sleep duration might cause hyperuricemia. In contrast, long sleep duration might be a protective factor. The findings suggest that maintaining healthy sleep habits would be a potential intervention to prevent hyperuricemia, which highlights the public health significance of sufficient sleep duration.

## Figures and Tables

**Figure 1 ijerph-19-08105-f001:**
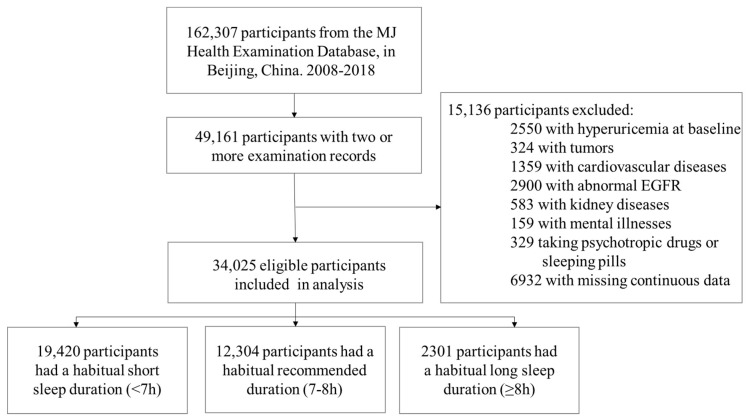
Flow chart for inclusion and exclusion criteria. EGFR indicates estimated glomerular filtration rate.

**Figure 2 ijerph-19-08105-f002:**
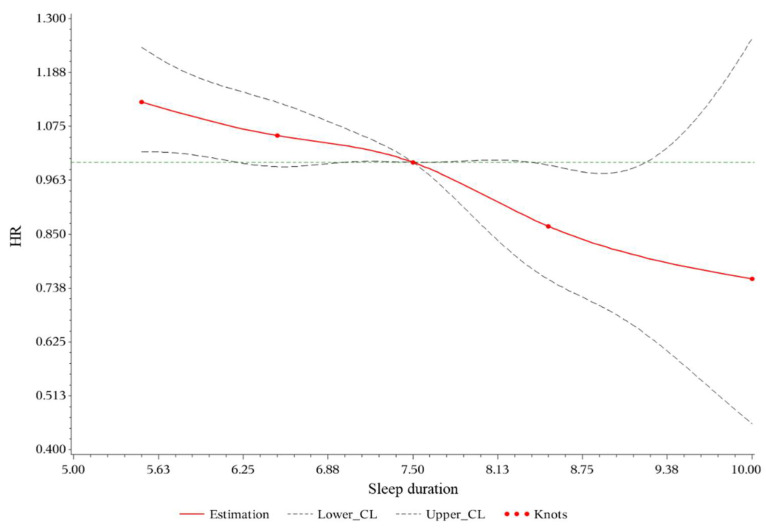
Restricted cubic spline plots for hyperuricemia by sleep duration. HR indicates hazard ratio; CL, confidence limit. Multivariable Cox regression models with restricted cubic splines show the hazard ratio for the incidence of hyperuricemia adjusted for sex, age, education level, personal income, marital status, smoking status, drinking status, exercise status, Body Mass Index, abdominal circumference, hypertension, diabetes, triglycerides, high-density lipoprotein cholesterol, and amount of white blood cells. The dotted lines indicate 95% CL. The x-axis represents the sleep duration. The y axis represents the HR of hyperuricemia.

**Table 1 ijerph-19-08105-t001:** Baseline characteristics of participants by sleep duration.

Characters	Missing	Habitual Sleep Duration			Overall
		<7 h	7–8 h	≥8 h	
Women (N, %)	0	10,087 (51.94)	6924 (56.27)	1491 (64.80)	18,502 (54.38)
Age (years, mean ± SD)	-	39.60 ± 9.83	36.59 ± 8.98	36.58 ± 8.97	38.31 ± 9.59
Married (*n*, %)	1588	14,870 (76.57)	9234 (75.05)	1754 (76.23)	25,858 (76.00)
School (*n*, %)	10,833				
Less than Master’s degree		1462 (7.53)	609 (4.95)	193 (8.39)	2264 (6.65)
Master’s degree		3594 (18.51)	1752 (14.24)	374 (16.25)	5720 (16.81)
Doctoral degree		8715 (44.88)	5482 (44.55)	1011 (43.94)	15,208 (44.70)
Annual income (*n*, %)	3466				
CNY < 100,000		9183 (47.29)	5644 (45.87)	1165 (50.63)	15,992 (47.00)
CNY 100,000–200,000		5126 (26.40)	3547 (28.83)	568 (24.68)	9241 (27.16)
CNY ≥ 200,000		3099 (15.96)	1891 (15.37)	336 (14.60)	5326 (15.65)
Smoke (*n*, %)	1010	6369 (32.80)	3053 (24.81)	584 (25.38)	10,006 (29.41)
Drink (*n*, %)	1161	4540 (23.38)	2191 (17.81)	395 (17.17)	7126 (20.94)
Habitual sports (*n*, %)	0	1042 (5.37)	184 (1.50)	0 (0.00)	1226 (3.60)
Overweight or obesity	-	4186 (34.02)	8223 (42.34)	720 (31.29)	13,129 (38.59)
Abdominal obesity (*n*, %)	50	4069 (20.95)	1784 (14.50)	325 (14.12)	6178 (18.16)
Hypertension (*n*, %)	18	2159 (11.12)	920 (7.48)	180 (7.82)	3259 (9.58)
Diabetes (*n*, %)	0	1015 (5.23)	398 (3.23)	80 (3.48)	1493 (4.39)
High level WBC	0	4405 (22.68)	2467 (20.05)	512 (22.25)	7384 (21.70)
Low HDL-C (*n*, %)	2798	2579 (13.28)	1460 (11.87)	233 (10.13)	4272 (12.56)
High TG (*n*, %)	26	3519 (18.12)	1740 (14.14)	353 (15.34)	5612 (16.49)

Abbreviation: CNY, Chinese Yuan; WBC, white blood cells; HDL-C, high-density lipoprotein cholesterol; TG indicates triglycerides. Note: *p* for difference of all characters was < 0.05. Overweight and obesity was defined as BMI ≥ 24.

**Table 2 ijerph-19-08105-t002:** Association of sleep duration with risk of hyperuricemia in different model.

Models	HR (95%CI)			*p* for Trend
	<7 h	7–8 h	≥8 h	
Events	2953	1668	247	
Person year	69,780.58	44,966.25	8511.92	
Incidence Rate (/1000 py)	42.32	37.09	29.02	
Model 1	1.108 (1.043–1.178)	ref	0.863 (0.755–0.986)	<0.001
Model 2	1.100 (1.034–1.169)	ref	0.851 (0.744–0.973)	<0.001
Model 3	1.070 (1.006–1.138)	ref	0.846 (0.740–0.968)	<0.001
Model 4	1.066 (1.002–1.133)	ref	0.844 (0.738–0.965)	<0.001

Abbreviation: HR indicates hazard ratio; CI, confidence interval; py, person-year. Model 1: Only adjusted for sex and age; Model 2: education level, personal income, marital status, smoking status, drinking status, and exercise status were additionally adjusted based on Model 1; Model 3: BMI, abdominal circumference, hypertension, diabetes, triglycerides, and high-density lipoprotein cholesterol, were further adjusted based on Model 2; Model 4: further adjusted amount of white blood cell based on Model 3.

**Table 3 ijerph-19-08105-t003:** Association of sleep duration with risk of hyperuricemia in different subgroups and interactions.

Subgroup	HR (95%CI) by Habitual Sleep Duration			*p* for Interaction
	<7 h	7–8 h	≥8 h	
Sex				
women	1.055 (0.941, 1.184)	ref	0.905 (0.729, 1.123)	0.220
men	1.067 (0.992, 1.147)	ref	0.824 (0.694, 0.978)	
Age				
≥50	0.930 (0.779, 1.113)	ref	0.688 (0.444, 1.065)	0.405
<50	1.084 (1.015, 1.157)	ref	0.874 (0.759, 1.006)	
Education				
Less than Master’s degree	0.947 (0.746, 1.202)	ref	0.652 (0.403, 1.054)	0.193
Master’s degree	0.950 (0.815, 1.106)	ref	0.767 (0.556, 1.059)	
Doctoral degree	1.062 (0.971, 1.163)	ref	0.772 (0.626, 0.952)	
Marital status				
Unmarried	1.005 (0.873, 1.157)	ref	1.119 (0.837, 1.495)	0.152
Married	1.100 (1.025, 1.181)	ref	0.796 (0.681, 0.932)	
Annual income				
CNY < 100,000	1.017 (0.929, 1.112)	ref	0.882 (0.734, 1.060)	0.440
CNY 100,000–200,000	1.114 (0.990, 1.254)	ref	0.975 (0.750, 1.267)	
CNY ≥ 200,000	1.148 (0.987, 1.336)	ref	0.653 (0.447, 0.954)	
Smoking				
Never	1.087 (1.003, 1.178)	ref	0.845 (0.711, 1.005)	0.709
Current and ever	1.038 (0.941, 1.146)	ref	0.866 (0.696, 1.077)	
Drinking				
Never	1.080 (1.003, 1.164)	ref	0.904 (0.773, 1.058)	0.522
Current and ever	1.034 (0.923, 1.158)	ref	0.708 (0.540, 0.930)	
BMI				
≥24	1.005 (0.928, 1.089)	ref	0.801 (0.666, 0.963)	0.035
<24	1.163 (1.056, 1.280)	ref	0.923 (0.759, 1.122)	
Abdominal obesity				
No	1.070 (0.993, 1.151)	ref	0.881 (0.754, 1.029)	0.624
Yes	1.061 (0.950, 1.187)	ref	0.764 (0.585, 0.996)	
Hypertension				
No	1.102 (1.031, 1.177)	ref	0.878 (0.760, 1.013)	0.274
Yes	0.878 (0.744, 1.036)	ref	0.664 (0.457, 0.965)	
Low HDL-C				
No	1.006 (0.935, 1.083)	ref	0.824 (0.703, 0.967)	0.156
Yes	1.216 (1.061, 1.394)	ref	0.815 (0.588, 1.130)	
High TG				
No	0.996 (0.927, 1.070)	ref	0.813 (0.693, 0.955)	0.053
Yes	1.271 (1.129, 1.431)	ref	0.964 (0.755, 1.231)	
Diabetes				
No	1.071 (1.006, 1.141)	ref	0.859 (0.749, 0.985)	0.753
Yes	1.076 (0.811, 1.426)	ref	0.747 (0.394, 1.418)	

Abbreviation: HR indicates hazard ratio; CI, confidence interval; CNY, Chinese Yuan; BMI, Body Mass Index; HDL-C, high-density lipoprotein cholesterol; TG indicates triglycerides. All results were based on the fully adjusted model (Model 3). The results of subgroup analysis by habitual sports were not shown due to the low percentage (*n* = 1226, 3.60%) of participants who habitually take exercises at leisure time.

## Data Availability

The datasets analyzed during the current study are not publicly available due to the protection of privacy considering the ethics but are available from the corresponding author on reasonable request.

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
