# Peer review of "Association of Sleep Duration with Hyperuricemia in Chinese Adults: A Prospective Longitudinal Study"

_ijerph, 2022, doi:10.3390/ijerph19138105_

Round 1

Reviewer 1 Report

Overall a well designed study with implications for public health related to improved sleep.

In first paragraph please review for minor errors in language.

Please clarify in Discussion how covariates likely impacted model estimates related to Figure 2 e.g., A to D

In Discussion please offer brief discussion on likely role of BMI related to sleep duration and hyperuricemia.

Author Response

Dear editors and reviewer 1,

Thank you so much for your precious comments and advice concerning our manuscript entitled “Association of Sleep Duration with Hyperuricemia in Chinese Adults: A Prospective Longitudinal Study”. The comments are very valuable and helpful. We have read the comments carefully and uploaded files of the revised manuscript. Revisions in the text are highlighted in ‘track changes’ and the responses to the comments are as follows.

Reviewer 1:

Q1: In first paragraph please review for minor errors in language.

A1: Thank you for your advice. We apologize for the problem with language. We have checked and revised the first paragraph. (lines 36-49)

Q2: Please clarify in Discussion how covariates likely impacted model estimates related to Figure 2 e.g., A to D

A2: We are grateful for your suggestion. In Table 2, we presented the HR estimates in the stepwise adjusted models, which show the impact of covariates. In Figure 2, we further show the dose-response relationship in a graphical way. To avoid repeating similar information in both table and figure, we only left the fully adjusted restricted cubic spline plot in Figure 2. In the discussion section, we highlighted sleep as an independent risk factor for hyperuricemia. More details can be found in lines 234-236.

Q3: In Discussion please offer brief discussion on likely role of BMI related to sleep duration and hyperuricemia.

A3: Thank you for your precious advice. We discussed the role of BMI related to sleep duration and hyperuricemia in lines 263-268. Short sleep duration is associated with a 16% increase in risk of hyperuricemia in the participants with a BMI < 24. The interaction between sleep duration and BMI provides insights into screening high-risk populations and protection of susceptible people.

Moreover, according to the Academic Editor Notes, we have revised 1) the direction of arrowhead for exclusion in Figure 1, and 2) we used “CNY” instead of "RMB" and spell it out in the first appearance (lines 104-105, 169, 213-214, Table 1, and Table 3). Please do not hesitate to contact me if you have any further questions.

Reviewer 2 Report

Dear Editor,

I carefully read the manuscript by Yu et al.

My comments and suggestions are the following:

 - English language needs to be carefully revised.

 - Lines 36, 37: The references should be reported at the end of the sentence (before the stop).

 - In the bakground, the authors should more datailed the rationale of their analysis.

 - Lines 80, 88: Were the used questionnaires validated?

 - Lines 105-112: The authors should include references to the relevant guidelines they referred to.

 - All the abbreviations used in the manuscript should be defined at their first occurrence.

 - Line 114: The authors should declare how was the normal distribution of the variables assessed.

 - Table 3: The authors should replace the word "gender" with "sex". As a matter of fact, sex is usually categorized as female or male but there is variation in the biological attributes that comprise sex and how those attributes are expressed. Gender refers to the socially constructed roles, behaviours, expressions and identities of girls, women, boys, men, and gender diverse people.

 - In table 3 (and in the mauscript too), the authors refer to "HDL" and "LDL" instead of "HDL-C" and "LDL-C". Actually, this is a big mistake because they are different things.

 - The authors should consider to refer to doi: 10.1097/HJH.0000000000001927 in their manuscript.

 - References have to be formatted following the instructions for the authors of the Journal.

Author Response

Dear editors and reviewer 2,

Thank you so much for your precious comments and advice concerning our manuscript entitled “Association of Sleep Duration with Hyperuricemia in Chinese Adults: A Prospective Longitudinal Study”. The comments are very valuable and helpful. We have read the comments carefully and uploaded files of the revised manuscript. Revisions in the text are highlighted in ‘track changes’ and the responses to the comments are as follows.

Reviewer 2:

Q1: English language needs to be carefully revised.

A1: Thank you for your advice. We have checked and revised the manuscript carefully.

Q2: Lines 36, 37: The references should be reported at the end of the sentence (before the stop).

A2: Thanks for the comments, we moved the references to the end of the sentence. (Lines 39-40)

Q3: In the background, the authors should more detailed the rationale of their analysis.

A3: Thank you so much for your precious advice. More detailed background and the rationale of the study were supplied in Lines 57-62.

Q4: Lines 80, 88: Were the used questionnaires validated?

A4: Thank you for your comments. The questionnaires have not been validated, but they are widely used in population studies to collect socio-demographic and lifestyle behavioral factors. The question about sleep duration in the questionnaire was “On average, how many hours do you sleep per day”, and there were five options for the question: (1) < 6 hours; (2) 6-7 hours; (3) 7-8 hours; (4) 8-9 hours; (5) ≥ 9 hours. We will further validate our questionnaires in the future. Similar questions were used and validated in previous studies [1, 2].

Q5: Lines 105-112: The authors should include references to the relevant guidelines they referred to.

A5: We agree with the comments and added references in lines 113-124.

Q6: All the abbreviations used in the manuscript should be defined at their first occurrence.

A6: We are grateful for your reminder. We have checked the abbreviations and defined them at their first occurrence in abstract, main text, figures, or tables.

Q7: Line 114: The authors should declare how was the normal distribution of the variables assessed.

A7: We are grateful for your comments. The normal distribution of continuous variables was tested using shapiro-Wilk normality test, and we added this answer in lines 127-128.

Q8: Table 3: The authors should replace the word "gender" with "sex". As a matter of fact, sex is usually categorized as female or male but there is variation in the biological attributes that comprise sex and how those attributes are expressed. Gender refers to the socially constructed roles, behaviours, expressions and identities of girls, women, boys, men, and gender diverse people.

A8: Thanks for your advice. We replaced "gender" with "sex" throughout the manuscript. (Lines 141, 188, 198, and Table 3)

Q9: In table 3 (and in the mauscript too), the authors refer to "HDL" and "LDL" instead of "HDL-C" and "LDL-C". Actually, this is a big mistake because they are different things.

A9: We thank the reviewer pointing the issue out. and sorry for the mistake. We corrected the word with “HDL-C” in the manuscript (lines 109-110, 120, 166, 169-170, 214, 232-233, Table 1, and Table 3).

Q10: The authors should consider to refer to doi: 10.1097/HJH.0000000000001927 in their manuscript.

A10: We are very grateful for your suggestion. The article (doi: 10.1097/HJH.0000000000001927) has been refered in the latest version of our manuscript to discuss the interactions among metabolic variates (lines 260-263. It is a well-designed study which found LDL-C and serum uric level were associated with incident hypertension, and indicated that complex interaction between LDL-C and serum uric level exists.

Q11: References have to be formatted following the instructions for the authors of the Journal.

A11: We apologize for the reference problem in the manuscript. Date of the reference has been revised to be bold, and Endnote was used to manage the references. (Lines 333-542)

  1. Liu, Y.; Wheaton, A. G.; Chapman, D. P.; Cunningham, T. J.; Lu, H.; Croft, J. B., Prevalence of Healthy Sleep Duration among Adults--United States, 2014. MMWR Morb Mortal Wkly Rep 2016, 65, (6), 137-41.
  2. Leng, Y.; Cappuccio, F. P.; Wainwright, N. W.; Surtees, P. G.; Luben, R.; Brayne, C.; Khaw, K. T., Sleep duration and risk of fatal and nonfatal stroke: a prospective study and meta-analysis. Neurology 2015, 84, (11), 1072-9.

Moreover, according to the Academic Editor Notes, we have revised 1) the direction of arrowhead for exclusion in Figure 1, and 2) we used “CNY” instead of "RMB" and spell it out in the first appearance (lines 104-105, 169, 213-214, Table 1, and Table 3). Please do not hesitate to contact me if you have any further questions.

Reviewer 3 Report

1.     The title clearly and precisely reflect the findings of the manuscript.

2.     The statistical methods are validate.

3.    The prior work is properly cited.

4.     I recommend accepting this manuscript for publication.

Author Response

Dear editors and reviewer 3,

Thank you so much for your precious comments and recommendation concerning our manuscript entitled “Association of Sleep Duration with Hyperuricemia in Chinese Adults: A Prospective Longitudinal Study”. We have read the comments carefully and we really appreciate your comments. Revisions in the text are highlighted in ‘track changes’ and the responses to the comments are as follows. Please do not hesitate to contact us if you have any further questions.

Moreover, according to the Academic Editor Notes, we have revised 1) the direction of arrowhead for exclusion in Figure 1, and 2) we used “CNY” instead of "RMB" and spell it out in the first appearance (lines 104-105, 169, 213-214, Table 1, and Table 3). Please do not hesitate to contact me if you have any further questions.

Round 2

Reviewer 2 Report

Dear Editor,

I carefully read the revised version of the manuscript that is significantly improved in comparison with the previous one. However, I warmly encourage the authors to include among the limitations of the study that the used questionnaires were not validated (this is indeed a relevant flaw of the study).

Author Response

Dear editors and reviewer 2,

Thank you again for your precious comments and kind advice concerning our manuscript entitled “Association of Sleep Duration with Hyperuricemia in Chinese Adults: A Prospective Longitudinal Study”. The comments are very valuable. We have read the comments in Round 2 carefully and uploaded files of the revised manuscript. Revisions in the text are highlighted in ‘track changes’ and the responses to the comments are as follows.

Q1: I carefully read the revised version of the manuscript that is significantly improved in comparison with the previous one. However, I warmly encourage the authors to include among the limitations of the study that the used questionnaires were not validated (this is indeed a relevant flaw of the study).

A1: Thank you so much for your careful review and precious advice. We have added this limitation in lines 281-283 of the revised manuscript. Please do not hesitate to contact us if you have any further questions.
